# Integrative Taxonomy of *Armeria* Taxa (Plumbaginaceae) Endemic to Sardinia and Corsica

**DOI:** 10.3390/plants12112229

**Published:** 2023-06-05

**Authors:** Manuel Tiburtini, Gianluigi Bacchetta, Marco Sarigu, Salvatore Cambria, Paolo Caputo, Daniele De Luca, Gianniantonio Domina, Alessia Turini, Lorenzo Peruzzi

**Affiliations:** 1PLANTSEED Lab, Department of Biology, University of Pisa, Via Derna 1, 56126 Pisa, Italy; a.turini10@studenti.unipi.it (A.T.); lorenzo.peruzzi@unipi.it (L.P.); 2Centre for Conservation of Biodiversity (CCB), Department of Life and Environmental Sciences, University of Cagliari, V.le S. Ignazio da Laconi 13, 09123 Cagliari, Italy; bacchet@unica.it (G.B.); msarigu@unica.it (M.S.); 3Department of Biological, Geological and Environmental Sciences, University of Catania, Via Antonino Longo 19, 95125 Catania, Italy; cambria_salvatore@yahoo.it; 4Department of Biology, University of Naples Federico II, Via Cinthia 26, 80100 Naples, Italy; pacaputo@unina.it (P.C.); daniele.deluca@unina.it (D.D.L.); 5Department of Agricultural, Food and Forest Sciences, University of Palermo, Viale delle Scienze 4, 90128 Palermo, Italy; gianniantonio.domina@unipa.it

**Keywords:** phylogeny, *Armeria*, integrative taxonomy, endemism

## Abstract

Sardinia and Corsica are two Mediterranean islands where the genus *Armeria* is represented by 11 taxa, 10 out of which are endemic. An integrative approach, using molecular phylogeny, karyology, and seed and plant morphometry was used to resolve the complex taxonomy and systematics in this group. We found that several taxa are no longer supported by newly produced data. Accordingly, we describe a new taxonomic hypothesis that only considers five species: *Armeria leucocephala* and *A. soleirolii*, endemic to Corsica, and *A. morisii*, *A. sardoa*, and *A. sulcitana*, endemic to Sardinia.

## 1. Introduction

Island biodiversity attracted the attention of many biologists since the mid-XIX century [1,2,3,4]. Islands have been historically classified according to their geological history into two large groups: oceanic and continental [5,6]. In the Mediterranean, among the largest (>5000 km^2^) islands—Sicily, Sardinia, Cyprus, Crete, and Corsica—only Cyprus can be considered an oceanic island [7], whereas all the others are continental islands. Speciation processes in the two types of islands are different [8,9,10].

Sardinia and Corsica share a similar Oligocenic–Miocenic (ca. 30 to 16 Ma) geological history [11,12] that played a role in shaping the current biodiversity. In fact, some authors consider them as a single biogeographic unit [13,14]). More recently, the Messinian salinity crisis (5.96–5.3 Mya) led to a high rate of extinction of the tertiary flora, followed by the onset of the Mediterranean climate [7].

It is generally assumed that continental islands should possess a higher rate of palaeoendemics. However, Cheikh Albassatneh et al. [15] failed to find any evidence supporting this idea for the flora endemic to Sardinia and Corsica. Irrespective of its origin, this flora shows a much higher rate of endemism than the mainland [16] due to isolation processes, reduced gene flow and adaptation, resulting in a higher rate of speciation [17]. The current floristic diversity of Sardinia and Corsica [18] is one of the highest in the Mediterranean. Indeed, 485 out of the 4572 plant species native to Sardinia and Corsica are endemic, and the Corso-Sardinian biogeographic province is currently considered a hotspot for plant diversity in the Mediterranean Basin [13,19,20,21].

Despite the Corsican-Sardinian flora is well known from a floristic point of view, the taxonomy of many endemic species is still unclear. One of the genera showing the highest number of endemic taxa for the region is certainly *Armeria* (DC.) Willd. (Plumbaginaceae). It is a genus of perennial, diploid (2*n* = 2*x* = 18) herbs, currently including 95 accepted species [22].

In Sardinia and Corsica, 11 *Armeria* taxa are reported, including the easily recognizable western Mediterranean *A. pungens* Hoffmanns. & Link, whose occurrence in the area could be explained by an event of long-distance dispersal from Portugal [23]. The other 10 taxa are all endemic to Sardinia and Corsica [24,25]. Arrigoni [24] considered all these endemics (except *A. morisii* Boiss.) as neoendemics, speculating about their origin and affinities with Sicily, Spain, and mainland Italy.

Speciation (and, in a broad sense, the rate of endemism) depends on our understanding of the meaning and the value of the taxonomic hypotheses that we assume as the outcome of the speciation events [26]. Accordingly, it is evident that systematics is a fundamental tool to unveil the distribution and boundaries of the variability of nature that we want to encapsulate in a name and a hierarchy using nomenclature and taxonomy [27]. However, the accuracy of systematic inferences depends upon the techniques used and the available resources [28]. Turrill [29] distinguished three levels of taxonomy. The first one is alpha-taxonomy, an approach that relies only on morphological data. When cytogenetic characters and/or anatomical characters are added, we can talk about beta-taxonomy. Lastly, omega-taxonomy refers to the ideal case of a taxonomic circumscription made using a high number of different sources of systematic information. Schlick-Steiner et al. [28] showed that when multiple sources of data are added (i.e., moving from alpha- to omega-taxonomy), the error rate in taxonomic circumscriptions steadily decreases. This is desirable, especially in a taxonomically difficult genus such as *Armeria*. Most of the species are still circumscribed on the basis of the sole morphology (i.e., at the alpha-taxonomic level). However, integrative taxonomic approaches (moving towards an omega-taxonomy) proved their effectiveness in resolving taxonomic uncertainties in *A. maritima* Willd. [30], *A. pubigera* Boiss. [31], *A. trianoi* Nieto Fel. [32], and *A. arenaria* (Pers.) F.Dietr. [33] or in detecting hybridization [34,35].

The latest taxonomic treatment of *Armeria* taxa endemic to Sardinia and Corsica dates back to Arrigoni [24], who relied only on morphology and chromosome counts. Thus, his taxonomic hypotheses could be potentially biased. Arrigoni [24] recognized six species and then four subspecies. For example, he circumscribed *A. multiceps* Wallr. subsp. *multiceps* and *A. sardoa* Spreng. subsp. *genargentea* Arrigoni as high elevational ecotypes, whereas he considered as low-elevational ecotypes *A. multiceps* subsp. *meridionalis* Arrigoni and *A. sardoa* subsp. *sardoa*. Similar infraspecific variations are described for *A. leucocephala* Salzm. ex W.D.J.Koch, endemic to Corsica, for which the author reported three subspecies [36]. Bernis [37] already highlighted morphological similarities among *A. leucocephala*, *A. multiceps*, and *A. soleirolii* (Duby) Godr. in Corsica. Doubts about the taxonomic value of the Corsican endemic taxa were already expressed by Tison and de Foucault [38]. Specifically, these authors consider the subspecies of *A. leucocephala* and *A. multiceps* as morphological variations at the individual level. Accordingly, moving towards an omega-taxonomic level for this group is crucial to reach a more stable taxonomy, which is an essential prerequisite for supporting conservation practices and clarifying the evolutionary processes that shaped the current diversity in Mediterranean islands.

Using an approach integrating molecular phylogeny, karyology, seed and plant morphometry, this study aims to (1) experimentally check the currently available taxonomic hypotheses (10 taxa, Figure 1) or alternatively, (2) propose a new taxonomic circumscription better fitting the newly produced systematic data with a new taxonomic scheme.

## 2. Results

### 2.1. Karyological Analysis

All populations were diploid showing 2*n* = 2*x* = 18 medium-sized (4.65 ± 0.81 μm) chromosomes. Centromeres were mostly median (50.21%), submedian (49.54%), and rarely subterminal (0.2%) (Appendix A). We occasionally observed terminal satellites on short arms (9 out of 98 metaphasic plates). No significant difference (at α = 0.01 level) was found among the currently accepted taxa. Selected plates are shown in Figure 2.

### 2.2. Molecular Phylogenetic Analysis

The results of the ILD test are shown in Appendix A and allow us to perform the concatenation of plastidial and nuclear markers. In total, the concatenated matrix contained 2384 characters: the nuclear marker (ITS) showed only seven polymorphic sites, whereas the plastidial markers showed 51 polymorphic sites (*trnH-psbA*: 12; *trnL-rpl32*: 28; *trnL-trnF*: 5; *trnQ-rps16*: 6), the ITS and plastidial trees are shown in Appendix A. The phylogenetic tree (Figure 3) shows a rather complex scenario, with many evolutionary grades and admixtures between populations and even individuals. The chosen outgroup is formed by two populations of *A. pungens*, whereas the other branches of the tree are mostly collapsed. *Armeria leucocephala* and *A. multiceps* populations form together a scattered paraphyletic group. One population, BI, is found in two completely different grades. Concerning *A. sardoa* Spreng. subsp. *sardoa* and *A. sardoa* subsp. *genargentea* Arrigoni, they form a well-supported clade (0.98) together; however, the latter subspecies is paraphyletic. Lastly, *A. morisii* and *A. sulcitana* Arrigoni from the type locality form a clade closely related to *A. soleirolii* and to part of the *A. leucocephala* accessions.

### 2.3. Seed Morphometric Analysis

According to Random Forest mean decreased accuracy, the five most important seed morphometric features are, in decreasing order: CHull, MBCRadius, Feret, Concavity, and MaxR (Appendix A). The overall accuracy of the final model on the current species circumscription was quite low (50.47%), albeit the AUC (Area Under the ROC Curve) was 0.859. The highest per class errors were found among the subspecies of *Armeria leucocephala* and *A. multiceps* (class error >66%) (Appendix A). On the contrary, the lowest values of misclassification were obtained by *A. sardoa* subsp. *sardoa* and *A. morisii* (27.09% and 30.1%, respectively). According to the tuned HDBSCAN* model, the whole dataset contains only two true clusters (Appendix A) for values ranging from 20 to almost zero, obtaining cluster scores (i.e., the sum of the stability scores for each salient flat cluster) of 59.40 (first cluster) and 315.54 (second cluster). The first cluster contains all the seeds from the GO and FO populations pertaining to *A. sulcitana* and *A. sardoa* subsp. *genargentea*, respectively. These two populations show much lower values for the five most important morphometric features found by Random Forest with respect to all others (Appendix A).

The second cluster contains all the remaining populations. A finer structure is found within this latter cluster, given that two unsupported subclusters can be observed. PCoA applied to the same dataset confirmed this pattern. The first two axes captured more than 86% of the total variability of the distance matrix (Appendix A). The KMO test returned a MSA = 0.5.

### 2.4. Plant Morphometric Analysis

The two first axes of the PCoA explain 58.4% of the total variance (Figure 4). All of the populations largely overlap in the morphometric space. Both *A. leucocephala* and *A. multiceps* show a large overlap among their respective subspecies, whereas some degree of separation is present between *A. sardoa* subsp. *sardoa* and *A. sardoa* subsp. *genargentea*, especially concerning the populations from Badde Urbara (BU), Monte Limbara (ML), and Aritzo (AR). *Armeria morisii* (TH) shows a clear separation from all other taxa (Figure 4 and Appendix A). The overall pattern is maintained when comparing the first and the third axis, but *A. soleirolii* shows an almost complete separation in the morphospace (Appendix A).

The best *k* value for kNN when applied to the current taxonomic hypothesis (Appendix A) at species level was *k* = 3. The multiclass AUC was 0.958, whereas the balanced accuracy was 92.8%.

The lowest values of balanced accuracy were obtained by the three subspecies within *A. leucocephala* (*A. leucocephala* subsp. *breviaristata* 69.15%; *A. leucocephala* subsp. *leucocephala* 60%; *A. leucocephala* subsp. *pubescens* 89.43%) followed by *A. multiceps* subsp. *multiceps* (88.6%). Moving to the univariate data, a heatmap (Figure 5) shows that the highest number of pairwise significant differences occurs between *A. sardoa* subsp. *genargentea* and *A. morisii* (29), whereas there is no difference between the subspecies within *A. leucocephala* and *A. multiceps*. Again, *A. morisii* shows the highest number of pairwise differences, followed by *A. sardoa* subsp. *genargentea* (135) and *A. sardoa* subsp. *sardoa* (104). In general, all subspecies within *A. multiceps* and *A. leucocephala* show <50 pairwise differences.

There is a high similarity among taxa, especially *A. sulcitana*, *A. leucocephala* and its subspecies, *A. multiceps* and its subspecies, *A. sardoa* and its subspecies. Accordingly, we tested two alternative taxonomic hypotheses based on the phylogenetic results and on morphometric comparisons. Hypothesis 1 consists in just lumping the subspecies under their respective species, while hypothesis 2 consists in also lumping *A. leucocephala* and *A. multiceps*. A tuned kNN model under hypothesis 1 showed a balanced accuracy of 97.87% with *k* = 5 (Appendix A), whereas a tuned kNN model for hypothesis 2 yielded a final balanced accuracy of 99.01% with *k* = 4 and AUC = 0.994 (Appendix A). In the Canonical Variate Analysis morphospace, where individuals were grouped according to hypothesis 2, *A. sardoa* is distinct from *A. sulcitana*, *A. leucocephala*, and *A. soleirolii* following the first and second discriminant functions, whereas *A. sulcitana* is distinct from other taxa when the first and third discriminant functions are plotted. These distinctions are confirmed by the lack of overlap of the 95% confidence ellipses (Appendix A).

## 3. Discussion

The currently accepted taxonomic hypothesis [24] is no longer supported by the new evidence produced in this study. Specifically, neither morphological nor molecular or karyological data is able to support the grouping hypothesis currently accepted for *Armeria* in Sardinia and Corsica (Figure 6).

Seed shape and size are under natural selection and may play a role in the adaptation of the population to different ecological conditions, leading to genetically fixing some characters and shaping the evolution of a species [39,40]. However, this is not the case for the *Armeria* taxa endemic to Sardinia and Corsica, in which we highlighted a high homogeneity in seed features. From a karyological point of view, there is no significant difference among taxonomic units, either arranging the taxa according to the current hypothesis or according to the alternative grouping hypotheses 1 or 2. Accordingly, there is a high degree of karyological homogeneity, suggesting that evolution in Sardinia and Corsica did not imply any relevant chromosomal rearrangement.

It is well known that homoploid hybrid speciation is frequent in *Armeria* [41,42]. Indeed, the karyotype is very stable with 2*x* = 2*n* = 18 [24,43], as also confirmed by our data, despite in other cases slight differences in karyotype structure can be observed [33]. The overall karyological homogeneity and the reduced prezygotic barriers, limited to the sole pollen-stigma dimorphism [44,45,46], may have facilitated the hybridization/introgression and reticulate evolution of these plants in Sardinia and Corsica [25,47,48]. Indeed, hybridization is common in *Armeria* [34,49], despite the low signal provided by ITS (Appendix A), which may be due to concerted evolution [47]. Accordingly, the phylogenetic signal found in the concatenated tree is largely dominated by the information coming from the cpDNA (Appendix A), which in *Armeria* is maternally inherited [50]. Hybridization/introgression, incomplete lineage sorting and concerted evolution could all explain the observed pattern in the *Armeria* clade containing all the taxa endemic to Sardinia and Corsica. There, some individuals belonging to the same population are sometimes scattered all over the tree (i.e., GO, MO, BI). Genomic data and more sophisticated models, such as coalescent network-like models, may be needed to disentangle among these possible events [51].

Nevertheless, we can see that the subspecies of *A. leucocephala* and *A. multiceps* are mixed and form a grade. This evidence is corroborated by the low number of univariate morphological features that separate these taxa, and supports the view by Tison & de Foucault [38], who did not recognize the taxonomic value of what they considered just altitudinal variations. Moreover, there is no possibility to morphologically separate neither the two species in their traditional concept, nor the two clades found by molecular phylogeny (Figure 6). Accordingly, we do think that the only sound taxonomic solution is to merge *A. leucocephala* and *A. multiceps* into a single species. In this respect, the name *Armeria leucocephala* Salzm. ex W.D.J.Koch has priority over *A. multiceps* Wallr. Concerning *A. sardoa*, *A. sardoa* subsp. *sardoa* is phylogenetically included within *A. sardoa* subsp. *genargentea*. Morphologically, the two subspecies show only six statistically significant differences, possibly due to adaptation to different altitudes. Accordingly, we deem more opportune not to distinguish subspecies within *A. sardoa*. *Armeria soleirolii* is narrowly endemic to the northwestern part of the rocky coast of Corsica, ranging from Calvi to Galeria [24]. Interestingly, this taxon is phylogenetically isolated within a grade composed of taxa both from Sardinia and Corsica. The unique habitat and morphological features (i.e., the presence of papillate epidermal cells on leaves that are also canaliculate and stiff) support the autonomy of this taxon, which indeed shows no misclassification by kNN. Lastly, *A. morisii* shows the highest number of unique morphological features. *Armeria morisii* and *A. soleirolii* are easily distinguished from all other taxa by a set of character states, and their position in the phylogenetic tree is better defined than in other cases. Our findings are partially in contrast with the observations published by Bernis [37], particularly for *A. soleirolii*. Finally, *A. sulcitana* is different from *A. sardoa* in having longer inner involucral bracts, longer awns and being taller (Appendix A), whereas it is slightly different from *A. leucocephala* in possessing mainly dimorphic leaves with exclusively smooth margins (Appendix A). Accordingly, considering the currently available systematic data together (Figure 6), we deem the alternative taxonomic hypothesis 2 (i.e., five species) the best possible solution. Thus, we can conclude that there was an overestimation of plant diversity for this genus in the biogeographical region of Sardinia and Corsica. Interestingly, this is not an isolated case. Indeed, other studies that used an integrative taxonomic approach focusing on taxa endemic to this region found a similar pattern in *Santolina* L. (Asteraceae) and other taxa ([52] and literature cited therein). A similar reduction in the number of taxa following integrative taxonomic approaches has been documented in *Ophrys* L. (Orchidaceae) [53] and *Pulmonaria* L. (Boraginaceae) [54] from elsewhere.

To facilitate the identification of the five species endemic to Sardinia and Corsica as newly circumscribed here (plus *A. pungens*), their morphological differences are summarized in an identification key (Section 4.1).

## 4. Taxonomic Setting

***Armeria leucocephala*** Salzm. ex W.D.J.Koch, Flora 6: 712. 1823.

Type (neotype designated here): FRANCE. Corsica: Monte d’Oro, s.d., *Salzmann* (G00440097 photo! https://www.ville-ge.ch/musinfo/bd/cjb/chg/adetail.php?id=298304 (accessed on 31 December 2022)).

Note: Koch [55] provided a description in German, also citing a collector (“*Salzmann*”) and the provenance “*in den Corsischen*”. Herbarium and types of Johann Friedrich Wilhelm Koch are unknown [56]. We found two herbarium specimens, one in G and one in MPU, where Salzmann’s herbarium is preserved [56,57]. The sheet barcode G00440097 bears two plants (part of the same collection) and the labels (bottom-left corner of the sheet) “*Armeria leucocephala Koch v. alpina* [manu Boissier]” and “*Statice leucocephala Koch Bot. Beit. 1823, m. Salzmann, Monte d’oro (Corsica)* [manu Salzmann]”. The sheet barcode MPU828624 bears five plants (part of the same collection), and the labels (bottom-left corner of the sheet) “*Statice | Montagnes de Corse*” and “*Statice leucocephala| Armeria leucocephala Koch in … | Montagnes de Corse*” (Salzmann’s handwriting; C. Loup pers. comm.). Unfortunately, both specimens lack the date of collection. In many sheets, Salzmann did not annotate the date (C. Loup pers. comm.). Despite this, it seems prudential to designate the specimen G00440097 as neotype of the name *Armeria leucocephala*. This specimen corresponds to the protologue and to the application of this name in this study.

= *Armeria leucocephala* subsp. *pubescens* (Salis) Arrigoni, Webbia 25: 160. 1970 ≡ *Statice armeria* var. *pubescens* Salis, Flora 17(2): 13. 1834.

Type (neotype designated here): FRANCE. Corsica: Monts di Cagna, pentes de Fontanella, Juin 1917, *P. Cousturier 2515* (P05063114 photo! https://science.mnhn.fr/institution/mnhn/collection/p/item/p05063114 (accessed on 31 December 2022)); isotypes: (P05093398 photo!; MPU1065617 photo!)

Note: Salis-Marschlins [58] described *Statice armeria* var. *pubescens* providing a Latin diagnosis and the following provenance in “Monte Cagna supra Portovecchio. 4000’ s. m.” No original material was found. Therefore, a neotype was designated with a specimen concurring with the diagnosis and coming from the type locally. This specimen agrees with the protologue and the data presented in this study and definitely allows us to consider this name as a heterotypic synonym of *A. leucocephala*.

= *Armeria multiceps* Wallr., Beitr. Bot.: 196. 1842.

Type (lectotype designated here): FRANCE. Corsica: Mt. Rotondo, 1828. *Soleirol* 3555 (P05386918 photo!, http://mediaphoto.mnhn.fr/media/1441385834889C0cZ4gyBPy7VI0iR (accessed on 31 December 2022)).

Note: In the protologue, Wallroth [59], reported “*Armeria montana* Soleir. In *herb. amic. Lucae*” and “*Auf dem monte d’oro in Corsica* (Eschenlohr.)”. We traced at P a specimen (barcode P05386918) bearing five plants and three labels, i.e., “*3555/Statice montana/Corse/Soleirol 1828”, “Soleirol, herb. Cors./3555. Statice montana/Mont. Rotondo*”, and “HERB. MUS. PARIS/*Armeria multiceps Wallr*.”. Both the collector (Soleirol) and the locality of the collection (“*Corse*” and “*Mont. Rotondo*”) match Wallroth’s protologue. The specimen collection date (1828) predates the protologue. This specimen is part of the original material for *A. multiceps*, and it is here designated as the lectotype. The lectotype agrees with protologue and the data presented in this study and definitively allowed us to consider this name as a heterotypic synonym of *A. leucocephala*.

= *Armeria leucocephala* subsp. *breviaristata* Arrigoni, Webbia 25: 159. 1970.

Type (holotype): FRANCE. Corsica: Monte Stello, rochers élevés, 1881, *Chabert* (FI!)

= *A. multiceps* Wallr. subsp. *meridionalis* Arrigoni, Webbia 25(1): 152 (1970)

Type (holotype): FRANCE. Corsica: Coldi Bavella: pascoli alberati a Sud con *Pinus laricio* m 1300–1450, substrato granitico, 1969, *P.V. Arrigoni* (FI!).

Other specimens seen: FRANCE, Corsica: Vette di M. d’Oro, 24 July 1907, *U. Martelli* (FI!)*; Massif du Monte Corvo, au S du Bocca di San Giuvanni (Col de Saint-lean) (Corse, dép. Haute-Corse, crête du Cap Corse à hauteur d’Ogliastro et de Sisco), alt. env. 980 m, rochers schisteux de la crête, avec *Anthyllis hermanniae*, *Hypochaeris robertia*, *Galium corsicum*, 4 June 1997, *J. Lambinon & G. Van Den Sande* (FI!); montis Cinto, Calacuccia 2000–2200 m, 15 July 1880, *E. Levier* (FI!); Localité Haute vallèe d’Asco riva gauche Cima della Statoja, roches des aretes silice 2200 m, 26 July 1906, *Voyage botanique en corse de E. Burnat, J. Briquet, A. Saint-Yves, F. Cavillier et E. Abrezol* (FI!); Capo Aggiorio près Evisa 1200 m, 31 May 1906; *Voyage botanique en corse de E. Burnat, J. Briquet, A. Saint-Yves, F. Cavillier et E. Abrezol* (FI!); s. loc., June 1849, July 1847, *Requien* (FI!); Corsica, July 1907, *U. Martelli* (FI!); Col di Bavella Isola Corsica 1200 m, 9 June 1914, *F. Spencer* (FI!); Rochers élevés, Monte Fosco au (*unclear word*) de S.ta Maria di Lota Corse, 12 July 1881, *A. Chabert* (FI!); Le Pigno, pic au-dessus de Farinola, 4 July 1866, *P. Marbille* (FI! *two specimens*); ibidem (PI!, three specimens); Corsica, 1881, *Groves* (FI!); Monte Renoso, July 1847, s. coll. (FI!); Corse Coscione, s.d., *Jordan* (FI!); Monte Coscione (Corsica), September 1933, *G. Massucci* (FI!); M.te Rotondo, in pascuis subalpinis inter ovilia “Del Timozzo” a Caccun (?) d’Argentis 18–1900 m, 11 July 1880, *E. Levier* (FI!); Vallum mayen de L’anghione prés de Vizzavona rochers silice 1300 m, 21 July 1906, *Voyage botanique en corse de E. Burnat, J. Briquet, A. Saint-Yves, F. Cavillier et E. Abrezol* (FI!); Paturages de la région alpine Mt. Rotondo Corse, 16 September 1880, *A. Chabert* (FI)!; lungo i torrenti nella foresta di Vizzavona, 7 July 1907, *U. Martelli* (FI!); M.te Rotondo, in rupestribus subalpinis iter vallis Restonica et ovilia “del Timozzo”, 11 July 1880, *E. Levier* (FI!); M.te Rotondo, in saxosis alpinis ad radices montis Muffrone supra lacum d’Argentu (*?*), 8 July 1880; *E. Levier* (FI!); Monte d’Oro versant W. panton rocheuses silice 2200, 11 Jun 1904, *F. Cavallier* (FI!)*; Corsica Vette del M.te d’Oro, 24 July 1907, *U. Martelli* (FI!); M.te d’Oro, 24 July 1907, *U. Martelli* (FI!)*; Vette di Monte Rotondo, July 1907, *U. Martelli* (FI!); Corsica Monte Rotondo, 30 July 1907, *U. Martelli* (FI!); Corsica Monte Rotondo, 30 July 1907, *U. Martelli* (FI!); Mont Incudine Graviers silice 1750–2130m, 18 July 1906, *Voyage botanique en corse de E. Burnat, J. Briquet, A. Saint-Yves, F. Cavillier et E. Abrezol* (FI!); Rupi alle vette del M.te d’Oro, July 1907, *U. Martelli* (FI!)*; Corsica, July 1907, *U. Martelli* (FI!); M.te rotondo, […] in rupestribus alpinis montis Muffrone supra lacum d’Argentu 2000m, 8 July 1880, *E. Levier* (FI!); M.te rotondo, in clivis saxosis supra lacum d’Argentu 2000m, 8 July 1880, *E. Levier* (FI!); Sommet du Rotondo, 2500 m, 2 August 1866, *P. Mabille* (FI!); Sommet du Rotondo, 2500 m, 2 August 1866, *P. Mabille* (FI!); Rochers au vettes de Bétotre (?) montagne de la Cagna (Corse), 10 June 1890, *Hefacci* (FI!)*; Gajou de rochers du la région alpine M. Rotondo Corse, 16 September 1880, *A. Chabert* (FI!); Bastélica Mont Renoso, 24 June 1878, *Reverchon* (FI!).

***Armeria morisii*** Boiss. in DC., Prodr. 12: 687. 1848.

Type (lectotype designated here):—ITALY. Sardinia, s.d., *Moris s.n.* (G00440188 photo!, http://www.ville-ge.ch/musinfo/bd/cjb/chg/adetail.php?id=300301&lang=en (accessed on 31 December 2022)).

Note: In the protologue, Boissier [60] provided a detailed description of the provenance and collector, adding “v.s.” [vidi specimen] and cited “*Armeria latifolia* Moris fl. sard. elench. fasc. 3, p. 10 non W. [Willdenow]” referring to *Stirpium sardoarum elenchus* by Moris (1827), where the author listed Willdenow’s *A. latifolia* and the locality “*inter rupes calcareas Oliena*”. In addition, Boissier [60] reported in synonymy “*Statice cephalotes* Bertol. fl. ital. v. 3, p. 511, quoad pl. sardoum” (see Bertoloni 1837, “*ex rupibus d’Oliena in Sardinia ab Eq. Prof. Morisio*”). At G, there is a specimen (barcode G00440188) bearing parts of the same plant and seven labels, of which one reports Moris’s annotation “*Armeria latifolia* Willd./*Sardinia/Moris*” and another one states “*Armeria morisii*! Boiss”. This specimen reports both Moris and Boissier handwritings and is clearly original material, here designated as the lectoype. It matches the original description and corresponds to the data and application of this name in this study.

***Armeria sardoa*** Spreng., Syst. Veg., ed. 16, 4(2): 127. 1827.

Type (neotype here designated): ITALY, Sardinia: Massif du Gennargentu, Monte Spada (Sardaigne, prov. Nuoro, extrémité nord du massif du Gennargentu), alt. 1240–1280 m, pentes granitiques à exposition ouest, lande herbeuse à *Genista corsica*, *Carlina macrocephala*, *Cerastium boissierianum*, 8 June 1981, *J. Lambinon 81/176 & J. Rousselle* (L2643496 Photo!; https://medialib.naturalis.nl/file/id/L.2643496/format/large (accessed on 31 December 2022)); isotype: (BR0000027279212V Photo!).

Note: Sprengel [61] described this species as an addition to those already listed in volume 1 of Sprengel’s *Systema vegetabilium* [62]. He provided a short diagnosis (“16. *A. [Armeria] foliis filiformibus canaliculatis scapoque glabris, foliolis involucri obtusis*”) and the two generic localities of Sardinia and Corsica. Plumbaginaceae of Sprengel’s herbarium were housed in B, but a large part of it was destroyed during the Second World War [56,63]. No original material was found in the main European herbaria consulted; therefore, a neotype was selected. This specimen corresponds to the protologue and to the data and application of this name in this study.

= *Armeria sardoa* subsp. *genargentea* Arrigoni, Webbia 25(1): 166. 1970.

Type (holotype): ITALY. Sardinia: da Bruncu Spina a P. Paolina e dintorni del Rifugio Lamarmora, 6 July 1969, *Arrigoni s.n.* (FI!).

***Armeria soleirolii*** (Duby) Godr. in Gren. & Godr., Fl. France 2: 737. 1853. ≡ *Statice soleirolii* Duby in DC., Bot. Gall. 2: 1032. 1830.

Type (lectotype here designated):—CORSICA: 3559 *Statice littoralis*? Calvi, Soleirol dedit 1828 (MPU021638 photo!).

Note: *Statice soleirolii* was published in Candolle’s *Botanicon Gallicum* (1830: 1032) [64] on the basis of an unpublished manuscript by Duby, as reported in the protologue before the diagnosis: “*S. soleirolii* (Dub. mss. [manuscript])”; provenance and collector were provided (“Ad rupes maritimas Corsicae circa Calvi (cl. Soleirol)”). We found four specimens in MPU and one in P collected in Calvi: P00326603, including the printed label “Soleirol, herb. cors., *Statice Soleirolii* Duby Bot. gall t. 2. p. 1032, H. prope Calvi.–Soleirol”; MPU021636 has the same label; MPU021637 bears the handwritten label “*Armeria Soleirollii* nob., *Statice Soleirollii* Duby bot. gall. suppl. p. 1032!, Mi Requien 1843, Corse Calvi juin 1822”; MPU021639 bears the handwritten label “3559 *Statice fasciculata* De Calvi, Corse, Soleirol” and MPU021638 bears the handwritten label “3559 *Statice littoralis*? Calvi, Soleirol dedit 1828”. The specimens P0032603, MPU021636 and MPU021639 lack the collection date. The specimen MPU021637 was collected before the publication of the name but did not include the name of the collector. Thus, we selected as lectotype the specimen MPU021638 because it is the only one including both the collection date and collector. It fits the original description and corresponds to the data and the application of this name in this study.

Other specimens seen: FRANCE, Corsica: Galeria (Corse, côte occidentale), rochers granitiques en bord de mer, a l’ouest du village, alt. env. 20–30 m, 3 June 1981, *J. Lambinon & J. Rousselle* (FI!)*; Corsica NW Isola Gargalu, 6 August 1975, *B. lanza & M. Borri* (FI!); Corsica NW Scoglio di Capo Mursetta, 5 August 1975, *B. lanza* (FI!); Corsica NW–Golfo di Girolata Scoglio di Soleiroil, 6 August 1975, *B. lanza & M. Borri* (FI!); Corsica–Calvi penisola de La Revellata rupi costiere granitiche, 3 June 1968, *P.V. Arrigoni & C. Ricceri* (FI!)*; Corsica–Calvi penisola de La Revellata rupi costiere granitiche, 3 June 1968, *P.V. Arrigoni & C. Ricceri* (FI!)*; Corsica–Calvi penisola de La Revellata rupi costiere granitiche, 8 September 1987, *P.V. Arrigoni, B. Corrias, S. Diana* (FI!)*; *Corse: Rochers maritimes à Calvi (1)*, 28–29 June 1937, *J. Chevalier* (FI!); Corsica–da Calvi a Porto Torre di Galeria, 9 September 1987, *P.V. Arrigoni, B. Corrias, S. Diana* (FI!); Corsica –Calvi Costa rocciosa in Loc. Portu Vecchiu Esp. SW, 20 September 1990, *P.V. Arrigoni, P.L. Di Tommaso, S. Diana* (FI!).

***Armeria sulcitana*** Arrigoni, Webbia 25(1): 169. 1970.

Type (holotype): ITALY. Sardinia: Com. Pula, Monte Santo, rocce di Punta sa Crescia, 3 July 1969, *P.V. Arrigoni s.n.* (FI!).

### 4.1. Identification Key to Armeria in Sardinia and Corsica

Before starting the identification, refer to the supplementary material published by Tiburtini et al. [33], where calyx and leaf apex characters are graphically exemplified. We also recommend taking several measures and computing the mean values of the mentioned characters. The key was built using herbarium material. Descriptive statistics and contingency tables for the newly circumscribed taxa are provided in Appendix A.

1.Interfloral bracts absent; involucral bracts leathery and ferruginous; calyx showing a long spur (>1 mm) at the base; plants of sand dunes …………………………. ***A. pungens***1.Interfloral bracts present; involucral bracts not leathery and light brown; calyx with indistinct spur; plants not from sand dunes ………………………………………………… 22.Leaves stiff and canaliculate with evidently papillate epidermal cells (Appendix A); flowers pink; width of the inner involucral bract 3.8 ± 0.5 mm …………… ***A. soleirolii***2.Leaves smooth, without evidently papillate epidermal cells; flowers pink or white; inner involucral bract 2.4 ± 0.9 mm ………………………………………………………… 33.Summer leaves oblong-lanceolate, 5 ± 1.31 mm wide, flat; flowers pink; intermediate involucral bract 10.1 ± 1.3 mm long……………………………………………. ***A. morisii***3.Summer leaves linear, <2.5 mm wide, flat or canaliculate; flower pink or white; intermediate involucral bract <8 mm long…………………………………………………… 44.Outer spikelet bracteole 5.0 ± 0.9 mm long; limb 2.38 ± 0.40 mm; inner spikelet bract 6.6 ± 1.2 mm long; leaves homomorphic.…………………………………… ***A. leucocephala***4.Outer spikelet bracteole 4.2 ± 0.7 mm long; limb 2 ± 0.56 mm; inner spikelet bract 5.7 ± 0.9 mm long; leaves mostly dimorphic…………………………………………………… 55.Leaf margin smooth (microscope!); inner involucral bract 3.1 ± 0.7 mm wide; calyx awn 0.8 ± 0.2 mm long; calyx pleurotrichous ………………………………… ***A. sulcitana***5.Leaf margin smooth or dentate (microscope!); inner involucral bract 1.8 ± 0.5 mm wide; calyx awn 0.4 ± 0.2 mm long; calyx holotrichous or pleurotrichous …………………………………………………………………………………………… ***A. sardoa***

## 5. Materials and Methods

### 5.1. Sampling

We selected 15 populations (Table 1) across Sardinia and Corsica. The populations studied were selected based on three criteria: (1) to include all the type localities of the taxa considered in this study (BS, CB, MCA, ML, MO, MR, MS, RE, SP, and TH—acronyms as in Table 1); (2) to include other populations explicitly cited by Arrigoni [27]: AR, BU, GO, and BI; (3) to include other populations for those taxa showing a distribution range relatively larger than others (FO).

In total, we studied 197 herbarium specimens stored either in FI or in PI. Digitized specimens stored in the Herbarium Horti Botanici Pisani (PI) are freely available for consultation at http://erbario.unipi.it/ (codes in Table 1), whereas herbarium specimens stored in FI are listed in the *specimina visa* and marked with an asterisk (*). Concerning molecular systematics, dried leaves were collected from a subset of three individuals for each population and put in a paper bag with silica gel. Unfortunately, specimens from the BI population were unavailable for the morphometric analysis and were used only for the phylogenetic reconstruction. Ripen fruits were also collected from the same populations. The seeds were dried, cleaned, selected, and stored at −25 °C in the Sardinian Germplasm Bank (BG-SAR), Cagliari, Italy.

**Table 1 plants-12-02229-t001:** Taxa and populations of *Armeria* taxa endemic to Sardinia and Corsica sampled in this study, according to the currently accepted taxonomic hypothesis [24]. * = type locality; (*n*) = specimens used for plant morphometry.

Current Taxonomic Hypothesis	Code	Population	Voucher	(n)
*A. leucocephala* Salzm. ex W.D.J.Koch subsp. *breviaristata* Arrigoni	MS *	France, Corsica, Monte Stello (Brando) [WGS84: 42.7890 N, 9.418194 E], 1290 m	*G. Bacchetta*, *S. Cambria*, 23 June 2021 (PI 062283–062288)	6
*A. leucocephala* subsp. *leucocephala*	MR *	France, Corsica, Monte Renoso (Ghisoni) [WGS84: 42.074306 N, 9.148889 E], 1724 m	*G. Bacchetta*, *S. Cambria*, 23 July 2021 (PI 062281–062282); FI	5
*A. leucocephala* subsp. *pubescens* (Salis) Arrigoni	MCA *	France, Corsica, Monte Cagna (Sotta) [WGS84: 41.597000 N, 9.130611 E], 1061 m	*G. Bacchetta*, *S. Cambria*, 20 July 2021 (PI 062289–062292); FI	5
*A. leucocephala* subsp. *pubescens* (Salis) Arrigoni	BI	France, Corsica, Bocca Illarata (Zonza) [WGS84 41.702861 N, 9.210111 E], 967 m	*G. Bacchetta*, *S. Cambria*, 21 July 2021 (CAG 64/21)	0
*A. morisii* Boiss.	TH *	Italy, Sardinia, Monte Corrasi–Thutturreli (Oliena) [WGS84: 40.249189 N, 9.426167 E], 1200–1240 m	*G. Bacchetta*, *S. Cambria*, *P. De Giorgi*, *A. Giacò*, 17 June 2020 (PI 062293–062302)	10
*A. multiceps* Wallr. subsp. *meridionalis* Arrigoni	CB *	France, Corsica, Col di Bavella (Quenza) [WGS84: 41.798111 N, 9.218611 E], 1367 m	*G. Bacchetta*, *S. Cambria*, 22 July 2021 (PI 062424–062429!)	6
*A. multiceps* subsp. *multiceps*	MO *	France, Corsica, Monte d’Oro (Vivario) [WGS84: 42.121250 N, 9.107167 E], 1255 m	*G. Bacchetta*, *S. Cambria*, 22 July 2021 (PI 062275–062280!); FI	9
*A. sardoa* Spreng. subsp. *genargentea* Arrigoni	BS *	Italy, Sardinia, Bruncu Spina (Fonni) [WGS84: 40.014564 N, 9.189225 E], 1810–1825 m	*G. Bacchetta*, *S. Cambria*, *P. De Giorgi*, *A. Giacò*, 16 June 2020 (PI 062436–062450)	15
*A. sardoa* subsp. *genargentea*	FO	Italy, Sardinia, Monte Spada (Fonni) [WGS84: 40.058586 N, 9.291667 E], 1330–1370 m	*G. Bacchetta*, *S. Cambria*; 27 June 2021 (PI 062303–062322)	20
*A. sardoa* subsp. *sardoa*	ML *	Italy, Sardinia, Monte Limbara (Berchidda) [WGS84: 40.852222 N, 9.183611 E], 1300–1330 m	*G. Calvia*, 13 June 2021 (PI 062404–062423)	20
*A. sardoa* subsp. *sardoa*	BU	Italy, Sardinia, Badde Urbara (Santu Lussurgiu) [WGS84: 40.157342 N, 8.626297 E], 940–950 m	*G. Bacchetta*, *S. Cambria*, *P. De Giorgi*, *A. Giacò*, 18 June 2020 (PI 062323–062342)	20
*A. sardoa* subsp. *sardoa*	AR	Italy, Sardinia, Valico Sa Casa (Aritzo) [WGS84: 44.401250 N, 9.994250 E], 970–980 m	*G. Bacchetta*, *S. Cambria*, *P. De Giorgi*, *A. Giacò*; 15 June 2020 (PI 062344–062363)	20
*A. soleirolii* (Duby) Godr.	RE *	France, Corsica, La Revellata (Calvì) [WGS84: 42.581111 N, 8.720000 E], 50 m	*G. Bacchetta*, *S. Cambria*, 22 July 2021 (PI 062430–062435); FI!	15
*A. sulcitana* Arrigoni	SP *	Italy, Sardegna, Monte Santo (Pula) [WGS84: 39.028983 N, 8.908394 E]	*G. Bacchetta*, 30 May 2021 (PI 062384–062403)	20
*A. sulcitana*	GO	Italy, Sardinia, Genn’e Spina (Gonnosfanadiga) [WGS84: 39.445444 N, 8.657167 E]	*M. Porceddu*, *M. Sarigu*, 3 June 2021 (PI 062516–062535)	20

### 5.2. Karyological Data

We followed the protocol described by Tiburtini et al. [39] to obtain metaphasic plates. Seeds germinated easily in Petri dishes with agar-agar at 1%, without any pretreatment, in 4–6(8) days. Chromosomes were observed with a Leitz Diaplan microscope, and pictures were taken with a Leica MC-170HD camera through Leica LAS-EZ 3.0 imaging software. From all the pictures taken, at least four metaphase plates for each population were selected and measured. Chromosome number and karyological parameters such as THL (Total Haploid Length), M_CA_ (Mean Centromeric Asymmetry), CV_CL_ (Coefficient of Variation of the Chromosome Length), and CV_CI_ (Coefficient of Variation of the Centromeric Index) were obtained from each plate with MATO 1.1 (version 20210101) [65]. All the statistical analyses, excluding the molecular systematics, were conducted in RStudio IDE [66]. Data were managed using the *tidyverse* package (version 2.0.0) [67], and pairwise permutation tests from the *rcompanion* package(version 2.4.26) [68] were used to test statistical significance reporting only *p*-values corrected with Benjamini–Hochberg procedure [69] (BH step-up procedure) to control the False Discovery Rate (FDR) at level = 0.01.

### 5.3. DNA Extraction and Molecular Systematics

We followed the protocol described by Tiburtini et al. [33] for DNA extraction, primer selection for internal transcribed spacers (ITS1 + 5.8S + ITS2) and four chloroplast intergenic spacers (*trnF–trnL*, *trnH–psbA*, *trnL–rpl32*, *trnQ–rps16*), their amplification, sequencing, and data analysis. ITS sequences and chloroplast intergenic spacers were submitted to DDBJ (ITS: LC757034-LC757084; *trnF–trnL*: LC757325-LC757375; *psbA–trnH*: LC757085-LC757135; *trnL–rpl32*: LC757226-LC757273; *trnQ–rps16*: LC757274-LC757324). Sequences were visually inspected and aligned using the ClustalW algorithm Sequences were visually inspected and aligned using the ClustalW algorithm [70] implemented in BioEdit (version 7.2.5) [71] with the default values. An incongruence length difference (ILD) test was carried out in Nona (version 2.0) [72] as a daughter process of Winclada (version 1.00.08) [73] to test the putative incongruence of nuclear and chloroplast partitions prior to combination; default values were used for the analysis. A nucleotide evolution model was calculated for each of the five sequenced regions using jModelTest (version 2.1.10) [74], and the best-fitting model was chosen over the others using the Bayesian Information Criterion (BIC) [75]. Bayesian phylogenetic trees for concatenated and nuclear markers were inferred in MrBayes (version 3.2.6) [76] in two simultaneous, independent runs with the following settings: 2,000,000 generations of MCMC sampling every 2000 generations and four runs (three cold and one hot). For the cpDNA, convergence was reached in 3,000,000 generations sampled every 3000 generations. Convergence and mixing were evaluated in Tracer (version 1.7.2) [77]. The consensus Bayesian tree was visualized in FigTree (version 1.4.2) [78] as implemented in BioEdit (version 7.2.5) with the default values. The best evolution models were JC for ITS and F81 for chloroplast markers, except for *trnL-rpl32*, which was F81+G. We were unable to amplify this plastidial marker from *A. soleirolii*. The tree was rooted using *A. pungens* (BM: Italy, Sardinia, Badesi Mare, (Badesi) [WGS84 41.127786 N, 9.085444E], 4 m, *G. Calvia*, 27 May 2021, PI 062536-062555; PL: Italy, Sardinia, Lu Riu de li Saldi, (Aglienu) [WGS84 41.127786 N, 9.085444E], 1 m, *G. Calvia*, 27 May 2021, PI 062364–062383).

### 5.4. Seed Morphometric Analysis

Seeds were selected and prepared for image analysis. The seeds were dried at the Sardinian Germplasm Bank (BG-SAR) of Cagliari (Italy) based on established international protocols down to 3–5% of internal moisture content, which guarantees homogeneity and regularity in seed size and weight and allows an effective long-term conservation [79]. Digital images of 100 seeds for each accession were acquired using a flatbed scanner (Epson Perfection V550 photo) with a digital resolution of 1200 dpi. When an accession had fewer than 100 seeds, the analysis was carried out on the whole batch available. Seeds were randomly placed on the scanner tray so that they did not touch one another: each accession was scanned using a light blue background to avoid interference from environmental light. Images were processed using the software package ImageJ v. 1.53v [(http://rsb.info.nih.gov/ij) (accessed on 28 December 2022)], and the plugin Particles8, freely downloadable on the official website [(https://blog.bham.ac.uk/intellimic/g-landini-software/) (accessed on 28 December 2022)], was used to measure 26 morphometric features (Appendix A). The raw final matrix was 1445 × 26 variables. We explored, visualized, and preprocessed the data through centering and scaling using the *tidyverse* package [67]. We build a Random Forest model using the *caret* package (version 6.0-93) [80] with 750 trees with a depth of 3. To both maximize model performance and reduce the effect of correlation, we tuned the hyperparameter *mtry* for 1 to 20. The dataset was split at each iteration in 75% training and 25% testing, and within the training set, we cross-validated the training data using the LGOCV method with 10 individuals left out at each iteration and applied it to the testing data. We selected the model that maximized ROC-AUC values instead of accuracy as a model performance metric to avoid the accuracy paradox. From the tuned model (*mtry* = 15), we extracted the most important morpho-colorimetric variables and the confusion matrix. Finally, we applied HDBSCAN* (Hierarchical Density-Based Spatial Clustering of Applications with Noise) [81,82], an unsupervised clustering algorithm. Classical clustering techniques such as K-mean are limited by the fact that (1) the number of clusters must be known a priori, (2) each point, even outliers, must belong to a cluster, and lastly (3) they assume some known probability density function (PDF) that may have generated the observed data. HDBSCAN* is a non-parametric, density-based clustering algorithm that resolves these problems and can be used both as a nonhierarchical and hierarchical clustering algorithm. The Manhattan distance was computed from the standardized matrix and used as input to HDBSCAN* from the *dbscan* package (version 1.1-11) [83]. The sole hyperparameter of the model, *minPoints*, was tuned by fitting 49 models with *minPoints* ranging from 2 to 50 and selecting the model that had the least number of dimensions (26) plus one [83,84] and minimized both the number of clusters and the points considered as noise points. We picked the value of 30 *minPoints* as the best model. After testing for suitability to factor analysis with the KMO test, PCoA from *ape* package (version 5.6-2) [85] corrected with Cailliez correction [86] based on the same distance matrix was used to visualize both the nonhierarchical clustering and the membership probability calculated from the estimated PDF.

### 5.5. Plant Morphometric Analysis

In total, 49 qualitative and quantitative morphological characters (Table 2) were studied, with a resulting dataset of 192 individuals × 49 variables. Macroscopic measures were taken with a digital caliper (error ± 0.1 mm) under a Leica A60 stereomicroscope, while microscopic measurements were taken through bar-scaled pictures in Fiji 2.1.0 [87]. To have a more objective way to count the number of leaf veins, free-handed transversal sections of leaves were carried out. We considered as a “vein,” each fascicule composed of xylem and phloem surrounded by sclerenchyma. The anatomy of summer leaves was surveyed under a Leitz Diaplan light microscope at 40×.

We used the *tidyverse* package [67] for data exploration, preparation and description, while we used the nearZeroVar function from the *caret* [88] package (version 6.0-93) for preprocessing, removing zero and near-zero variance variables. We tested the suitability of the data for factor analysis with the Kaiser–Meyer–Olkin test (MSA = 0.89, *psych* (version 2.2.9) [89] and Bartlett sphericity test (*p* < 0.001, REdaS package (version 0.9.4) [90]. Both were performed successfully on the numeric correlate matrix. PCoA (Principal Coordinate Analysis) was performed using the *pcoa* function in the FD package (1.0-12.1) [91], while the Cailliez correction [86] was applied due to the violation of the triangle inequality (i.e., the matrix was not Euclidean). Lastly, 95% confidence ellipses were calculated assuming a *t*-distribution of individuals in a morphometric space using the function *stat_ellipse* of *ggplot2* package (version 3.4.2) at the species level for the Canonical Variable Analysis (CVA) [67,92]. Categorical data were binary encoded using the encode_binary function from the *cleandata* package (version 0.3.0) [93]. To test the alternative taxonomic hypotheses, we applied the kNN algorithm from the *caret* package (version 6.0-93) [88]. We decided to use kNN, a non-parametric, powerful [94] supervised machine learning model, because, for some populations, a low number of individuals was available. The model was tuned using the *caret* package for the sole hyperparameter *k* applying LOOCV for a range of *k* values from 1 to 20 (i.e., the maximum number of individuals per population) and retaining the model that maximizes the multiclass, cross-validated, balanced accuracy value. We followed the separating or merging scheme proposed by Xiong et al. [95] in case of overlapping regions in the morphospace between classes (i.e., a priori groups to be tested) or propose new alternative hypotheses. Each character was statistically tested, comparing each species. For all the quantitative characters, a pairwise permutation test was applied (10,000 permutations for each test), while for qualitative variables, Fisher’s Exact test was used. Both functions derive from the *rcompanion* package (version 2.4.26) [68]. We used the Benjamini & Yekutieli [96] procedure to control the family-wise error rate due to multiple testing setting the α at 0.01. We obtained a 10 × 10 matrix containing the number of characters showing statistically significant differences for each pair of taxa. Based on this matrix, we created a heatmap from the *plotly* package (version 2.4.26) [97]. To build the identification key, we used the combination of the most representative decision tree from the Random Forest tuned model (*mtry* = 5.64, 1000 trees) extracted using the *ReprTree* function of the *reprtree* package (version 0.6) [98] and the comparisons of the effect size of each character through Cohen’s D and mosaic plots for qualitative variables. We selected those characters that maximized the pairwise Cohen’s D values given the species or group species pairs.

### 5.6. Nomenclature

Information about currently accepted names, basionyms, and homotypic synonyms was obtained from the Euro+Med-Checklist [99] and Peruzzi et al. [100]. Information about the herbaria in which the original material of these names could be stored was derived from Stafleu and Cowan [56]. Accordingly, we digitally examined the following herbaria: BR, COI, G, K, L, MPU, and P (herbarium acronyms follow Thiers [101] in search of potential original material. This allowed us to typify the names *A. leucocephala*, *A. morisii*, *A. multiceps*, *A. sardoa*, *Statice armeria* var. *pubescens*, and *Statice soleirolii*.

## Figures and Tables

**Figure 1 plants-12-02229-f001:**
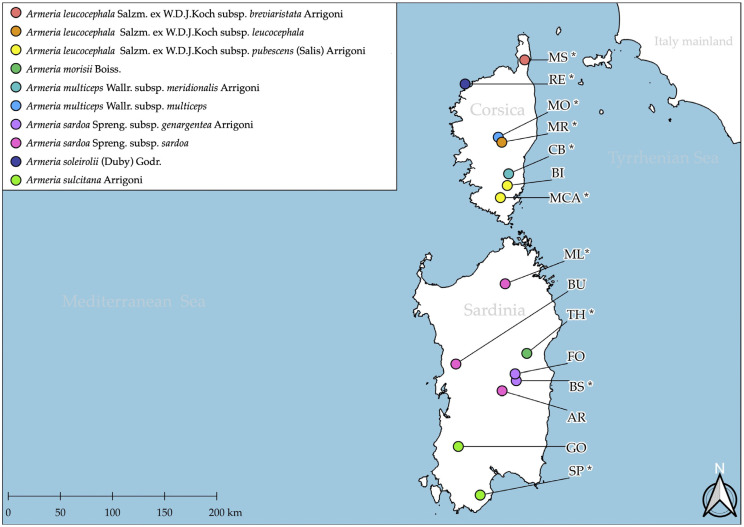
Distribution of the 10 taxa and 15 populations of *Armeria* taxa endemic to Sardinia and Corsica sampled in this study. Asterisk = type locality.

**Figure 2 plants-12-02229-f002:**
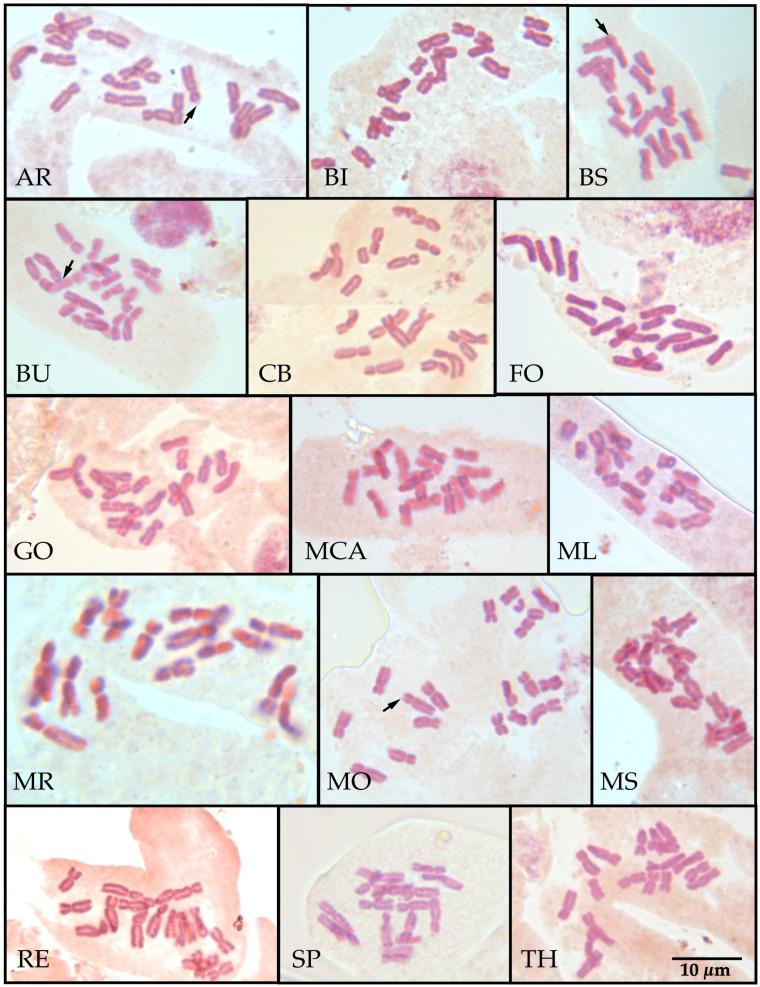
Selected metaphasic plates of the 15 populations of *Armeria* taxa endemic to Sardinia and Corsica considered in this study. Arrows indicate the presence of satellites. Population codes as in Table 1.

**Figure 3 plants-12-02229-f003:**
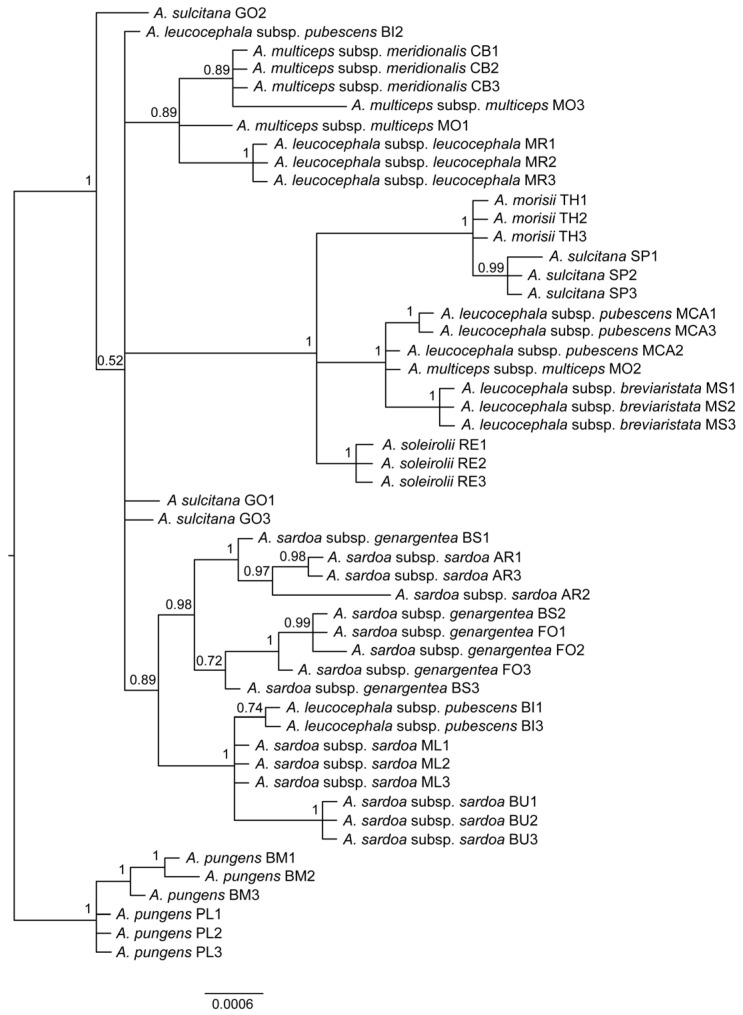
Phylogenetic tree of *Armeria* taxa endemic to Sardinia and Corsica based on the concatenated dataset of nuclear (ITS) and plastidial markers (*trnH-psbA*, *trnL-rpl32*, *trnL-trnF*, *trnQ-rps16*). Population acronyms are as in Figure 1.

**Figure 4 plants-12-02229-f004:**
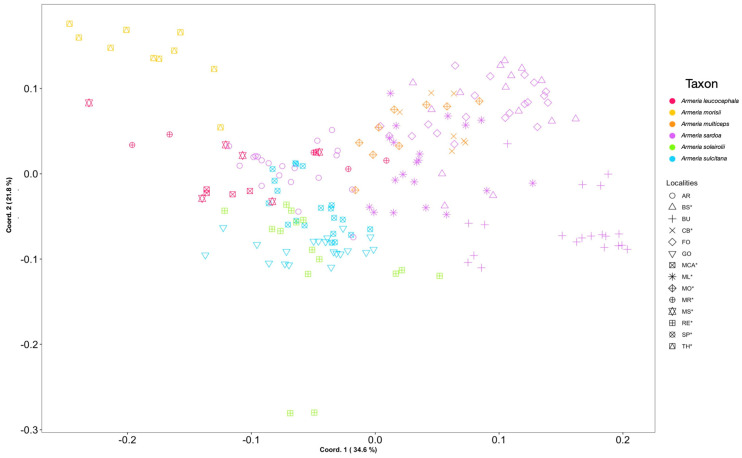
PCoA illustrating the morphometric variation in *Armeria* species endemic to Sardinia and Corsica based on the Gower distance of the 49 characters measured. Symbols indicate the population acronyms (Table 1), whereas the colors indicate the taxa at species level. Asterisk = type localities.

**Figure 5 plants-12-02229-f005:**
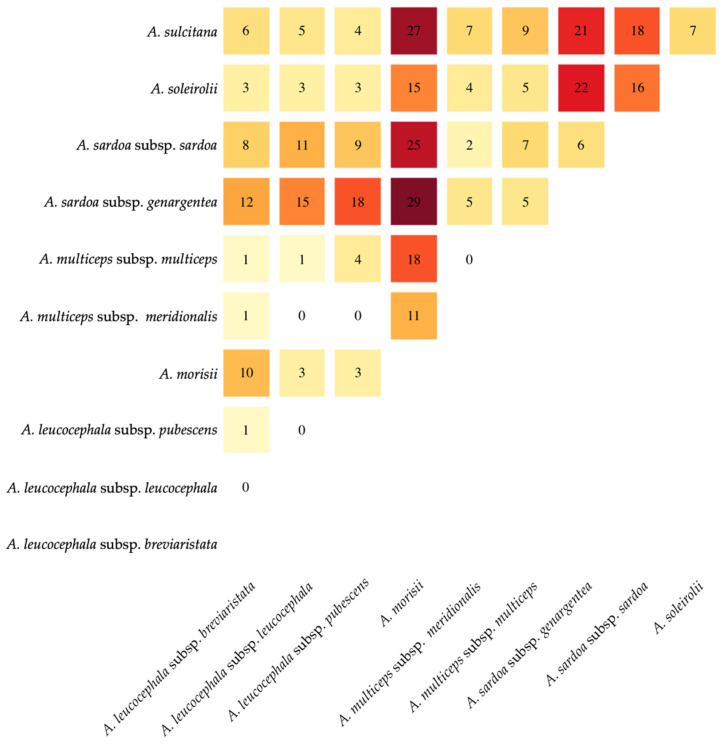
Heatmap summarizing univariate statistically significant pairwise differences in *Armeria* taxa endemic to Sardinia and Corsica based on the 49 characters measured. Color intensity is related to the number of characters showing significant differences.

**Figure 6 plants-12-02229-f006:**
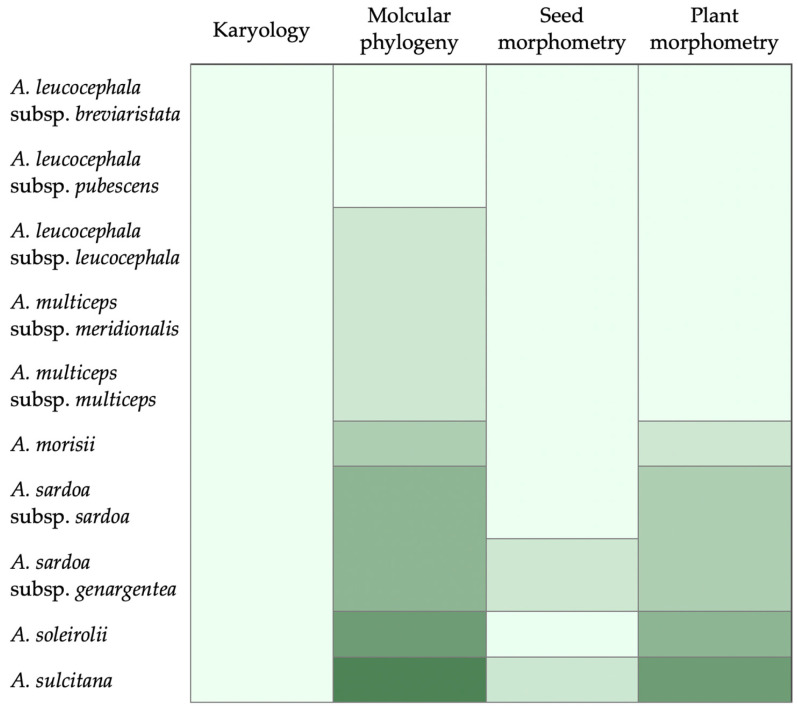
Schematic and simplified representation of the different lines of evidence available for the circumscription of the *Armeria* taxa endemic to Sardinia and Corsica. Different shades of green indicate different groups suggested by every single line of evidence.

**Table 2 plants-12-02229-t002:** Morphological characters used to study the *Armeria* taxa endemic to Sardinia and Corsica. QC = quantitative continuous, QD = quantitative discrete and, BI = binary.

Character Name	Description of the Character	Type	Tool
1. SPINE	Spine presence on the calyx awn (yes/no)	BI	Stereo
2. AWN	Awn presence on the calyx’s limb (yes/no)	BI	Stereo
3. PAP	Presence of papillae on leaves	BI	Stereo
4. CALYX_HAIRINESS	Calyx hairiness (holotrichous/pleurotrichous)	BI	Stereo
5. CALYX_VEINS	Number of calyx veins with hairs (10/5)	BI	Stereo
6. DIMORP	Leaf dimorphism (yes/no)	BI	Stereo
7. INNER_BRACT_HAIR	Presence of hairs on inner bract (yes/no)	BI	Stereo
8. MAR_SUM_LEAF	Margin of the summer leaf (hyaline/dentate)	BI	Stereo
9. MAR_WIN_LEAF	Margin of the winter leaf (hyaline/dentate)	BI	Stereo
10. OUTER_BRACT_HAIR	Presence of hairs on outer bract (yes/no)	BI	Stereo
11. SUM_LEAF_APEX	Shape of the summer leaf apex (acute/cucullate)	BI	Stereo
12. VEINS_HAIRS	Presence of hairs along the leaf veins (yes/no)	BI	Stereo
13. WIN_LEAF_APEX	Shape of the winter leaf apex (acute/cucullate)	BI	Stereo
14. COL_PET	Petal color (white/pink)	BI	Stereo
15. ANG_SUM_TIP	Summer leaf tip angle (°)	QC	Fiji
16. ANG_WIN_TIP	Winter leaf tip angle (°)	QC	Fiji
17. AWN_LENG	Awn length (mm)	QC	Fiji
18. DIAM_CAP	Capitulum diameter (mm)	QC	Calliper
19. HEIGTH	Plant height (mm)	QC	Ruler
20. LENG_CAL_PED	Calyx pedicel length (mm)	QC	Fiji
21. LENG_CAL_TUBE	Calyx tube length (mm)	QC	Fiji
22. LENG_INNER_INV_BRACT	Length of the involucral inner bract (mm)	QC	Calliper
23. LENG_INNER_SPI_BRACLE	Length of the inner spikelet bracteole (mm)	QC	Calliper
24. LENG_INNER_SPI_BRACT	Length of the inner spikelet bract (mm)	QC	Calliper
25. LENG_INTER_INV_BRACT	Length of the involucral intermediate bract (mm)	QC	Calliper
26. LENG_OUT_INV_BRACT	Length of the involucral outer bract (mm)	QC	Calliper
27. LENG_OUTER_SPI_BRACLE	Length of the outer spikelet bracteole (mm)	QC	Calliper
28. LENG_OUTER_SPI_BRACT	Length of the outer spikelet bract (mm)	QC	Calliper
29. LENG_SUM_LEAF	Summer leaf length (mm)	QC	Ruler
30. LENG_WIN_LEAF	Winter leaf length (mm)	QC	Ruler
31. LIMB_LENG	Limb length (mm)	QC	Fiji
32. SCA_DIAM	Scape diameter at 1 cm from the base (mm)	QC	Calliper
33. SCA_LENG	Scape length (mm)	QC	Ruler
34. SHEATH_LENG	Sheath length (mm)	QC	Calliper
35. WIDTH_CAL_TUBE	Width below the calyx tube limb (mm)	QC	Fiji
36. WIDTH_IAL_SUM	Width of the hyaline margin in summer leaf (mm)	QC	Fiji
37. WIDTH_IAL_WIN	Width of the hyaline margin in winter leaf (mm)	QC	Fiji
38. WIDTH_INNER_INV_BRACT	Width of the involucral inner bract (mm)	QC	Calliper
39. WIDTH_INNER_SPI_ BRACT	Width of the inner spikelet bract (mm)	QC	Calliper
40. WIDTH_INNER_SPI_BRACLE	Width of the inner spikelet bracteole (mm)	QC	Calliper
41. WIDTH_INTER_INV_BRACT	Width of the involucral intermediate bract (mm)	QC	Calliper
42. WIDTH_OUT_INV_BRACT	Width of the involucral outer bract (mm)	QC	Calliper
43. WIDTH_OUTER_SPI_ BRACT	Width of the outer spikelet bract (mm)	QC	Calliper
44. WIDTH_OUTER_SPI_BRATLE	Width of the outer spikelet bracteole (mm)	QC	Calliper
45. WIDTH_SUM_LEAF	Width of the summer leaf at the middle (mm)	QC	Calliper
46. WIDTH_WIN_LEAF	Width of the winter leaf at the middle (mm)	QC	Calliper
47. N_INV_BRACT	Number of involucral bract	QD	
48. N_SUM_VEINS	Number of veins (with sclerenchyma) of summer leaf	QD	Microscope
49. SCAP_NUM	Number of scapes	QD	

## Data Availability

R Scripts used to perform the data analyses are available on GitHub: https://github.com/Bendlexane/data-analyses-in-R/tree/main/Tiburtini%20et%20al.%2C%202023.

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
