# Peer review of "Integrative Taxonomy of Armeria Taxa (Plumbaginaceae) Endemic to Sardinia and Corsica"

_plants, 2023, doi:10.3390/plants12112229_

Round 1
Reviewer 1 Report
The ms. revises the taxonomy of Armeria in Corsica and Sardinia using an integrative approach that includes caryology, morphometry (seeds and other organs) and molecular phylogenetics. The purpose of the ms. is much justified since it is a complex genus which, due to the weakness of internal reproductive barriers, is prone to extensive gene flow, introgression and homoploid hybrid speciation, as the authors acknowledge in the discussion. But my point is that, in the context of their aim at moving towards an omega taxonomy, the processes responsible for the current diversity should be considered and, if relevant, explicitly addressed. My comments below aim at helping the authors improve the ms., which in my opinion requires revision along a few aspects, but deserves publication.
My biggest concern is the molecular phylogenetic part. The reader is given a phylogenetic tree based on Sanger sequences (ITS + four plastid non-coding regions) concatenated in a single matrix which, unless providing more information, raises doubts, because it does not properly consider the impact of hybridization in the phylogeny. The number of polymorphic sites in the plastid data (51) is seven times that in the ITS (the authors do not report which of those are parsimony informative sites). This situation is opposite to what you usually find in phylogenetic studies of most plant groups based on ITS and non-coding plastid sequences, where variation in ITS is higher than in cpDNA. The reason for the low number of low (7) ITS polymorphic sites in your Armeria samples was proposed years ago to be a combination of extensive gene flow and biased concerted evolution within territories (Fuertes et al. 1999, Syst. Biol.; Nieto Feliner et al. 2001, Cladistics). This combination of an organismal-level process (hybridization) with its impact on the genomic composition of a multicopy region (ITS) that undergoes homogenization across reproductive groups, apparently led to a geographic, rather than taxonomic, structure of variation, which was also found in other groups such as Centaurea (e.g., Suarez-Santiago et al., 2007, Mol.Phyl.Evol.). The four samples from Corsica and Sardinia sequenced in our 2003 study all fell within the same clade, which showed no internal resolution (Fuertes & Nieto Feliner, 2003, Mol.Phyl.Evol.). Therefore, it is not expected that an ITS phylogeny of exclusively Corsican and Sardinian samples is going to provide nucleotide variation that recovers signal of the species phylogeny, or any substantial ITS variation. The consequence is that the phylogeny you have generated has virtually only plastid signal which, obviously is also highly influenced by extensive hybridization. But the message that the reader gets (even if not explicitly stated in the ms.) is that the lack of incongruence detected in your ILD tests is due two different genomes (nuclear, plastome) recovering the same signal that corresponds to the species phylogeny. So, the implicit message from the phylogenetic part is, in my opinion, misleading.
What to do? I suggest that you explain the problem with the background of what we know in Armeria. I would present the two independent phylogenies. Then, I would explain the likely reasons why ITS variation is so meagre and conclude that you are spotting mostly plastid variation.
Other points:
Section Plagiobasis is not the correct name for all Armeria species not belonging in section Macrocentron. In De Candolle’s Prodromus, where the basyonym for A. sect. Plagiobasis is, A. maritima (the type species of the genus) is listed under this section. Besides, there’s no intraspecific division other than that separating Macrocentron from the rest. Because your ms. includes a formal taxonomic treatment, the nomenclature for infrageneric names should not be informal. An autonym (sect. Armeria) should likely be the correct name for sect. Plagiobasis.
The terminology for all types of bracts is inconsistent along the ms. You should use consistently the same terms for the same organs across the text, description of morphometric characters and identification key. Otherwise, it seriously compromises understanding even for somebody like me who has been looking at this genus for decades. For instance, (L. 60, 406, etc) you refer to “interfloral bracts”. Because spikelets frequently contain 3-5 flowers each, “interfloral bracts” could be thought of the bracteoles that sometimes occur inside spikelets in section Armeria. But you apply it apparently to spikelet bracts, i.e., those embracing the spikelets, which are absent in some species of section Macrocentron (including A. pungens). Certainly, the bracts integrating the involucre should be named “involucral bracts”, not scales, which leads to confusion when used as an alternative term for the same organ. Other authors have not been consistent in bract terminology for Armeria either. E.g., Arrigoni’s paper on the Italian peninsular Armeria (Fl. Medit. 2015), used “scales” (for involucral bracts) just in the figure captions. But, this aspect should be improved in your ms.
L. 188.- Figure 5 (heat map) is incorrectly numbered (“Fig. 4”) and cited (L. 180).
L. 225 and elsewhere.- “papillae on leaves”. Not clear what you mean. Epidermal cells are certainly convex, and the epidermis itself is thick as is in A. pungens. But, both in A. soleirolii and A. pungens, there are also small pits, which appear as dark spots in the old leaves. This correspond to Plumbaginaceae salt (Licopoli) glands.
L.230.- (diacritical) Do you mean diagnostic?
There are some wrong spellings or typos along the text (L. 129, L.181, L. 550, etc.). In general, it is well written.
Author Response
Dear Reviewer 1,
We would like to express our sincere gratitude for the invaluable tips and guidance you provided during the review process. Your insights and suggestions have been immensely valuable in improving the quality of our work. Thank you for taking the time to review my work and for your thoughtful feedback. Your contributions have undoubtedly made a significant impact on the quality of the ms.
We provided a point by point response to your review on our cover letter.
Sincerely,
The Authors

Reviewer 2 Report
The Ms reports an interesting study of integrative taxonomy for Armeria in Corsica and Sardinia. Armeria is a species rich genus in the Mediterranean which is challenging for taxonomy due to the incomplete reproductive isolation of almost all species. All data and results are relevant, and the study is an important progress for the study area and Armeria genus. There are no strong concerns but the presentation of results and the inferences leading to new taxonomical treatments need to be better explained.
Introduction:
The first paragraph is not useful because the issue is not a comparison of islands. More data on Corsica and Sardinia flora would more relevant, as well a short review of integrative taxonomy in Armeria genus. The sentences on “alpha”, “beta” and “omega” taxonomy are not very clear and need to be better explained if they are necessary.
Results:
The authors need to show and to analyse the rDNA ITS marker phylogeny and the plastid phylogeny separately. The ILD test is a very coarse and global test of concordance and not relevant to justify a concatenation of two types of markers having very different mode of heredity and evolution. For an integrative approach it is better to not merge them but to account for the different relationships, or similar ones, that they show.
The figure 3 is very confusing and weird, it should be explained because it is almost not understandable for the reader.
The chromosomes pictures should be in the main text. It is now very rare to have such data in paper, thus needs to be shown.
Concerning morphometry, the authors could use discriminant analysis to test the previous and new taxonomic treatment. Or more simply they could illustrate the ordination with the former and new taxonomic treatments. Showing the ITS and plastid clades could be interesting too. I do not understand why the authors performed a PCoA rather than a PCA. In the figure 4 the symbols and colors are too much faint and the graphic need to be improved.
Discussion :
The discussion is very short and did not clearly explain how the data support the new taxonomic treatment. A review of all the data taking the form of table could be very useful for the reader.
Author Response
Dear Reviewer 2,
We would like to extend our heartfelt appreciation for the invaluable tips and guidance you offered throughout the review process. Thank you for dedicating your time to reviewing our manuscript and providing thoughtful feedback. Your contributions have made a significant impact on the overall quality of the paper.
We have carefully addressed each point raised in your review in our cover letter attached.
Warm regards,
The Authors

Reviewer 3 Report
The presented manuscript is devoted to the taxonomic study of Armeria sect. Plagiobasis inhabiting Sardinia and Corsica. The authors applied the following methods: molecular phylogenetic study based on a concatenated data set of ITS and four plastid markers; karyological study; morphometric study of seeds; morphometric study of vegetative and generative plant characters. The authors presented valuable results concerning morphological and genetic variability in the studied group of species. However, the taxonomic conclusions drawn from these results raise many questions.
1. In the morphometric study, the hiatus among the majority of species is absent (Figure 4, PCoA results, coord. 1 & 2). The only species more or less isolated from the others is Armeria morisii. In the Fig. S5 (PCoA results, coord. 1 & 3), A. morisii takes a separate position and A. soleirolii can be more or less distinguished. On the Fig. S4 (Canonical Variate Analysis results), the species A. morisii is separated by hiatus. Two populations of Armeria sardoa (BS and AR) can be distinguished from the others but not grouped with each other and with other populations of A. sardoa. All other populations form a mixed common group. In Fig. S4 would be better to denote species rather than populations. Perhaps the authors should better select morphological features and include only those that have the maximum distinguishing ability. In general, it seems that morphological features poorly separate the species of the studied group, at least, this has not been demonstrated by multidimensional analyses.
2. A heat map summarizing one-dimensional statistically significant pairwise differences also demonstrates that A. morisii and A. sardoa are the most isolated from other species. However, if the studied species were well isolated, this would be reflected in the multivariate analysis, but this is not the case. In the Identification key we also see that many characters have overlapping ranges.
3. The majority of studied species are not monophyletic according to molecular phylogenetic data (Figure 2). The only species which form a monophyletic clade is A. soleirolii. A. sardoa also forms more or less supported clade, but the clade is not monospecific and includes also two accessions of A. leucocephala subsp. pubescens.
4. It is unclear whether the Plagiobasis section is a monophyletic group or not. If the section is not monophyletic, then the study of species from Sardinia and Corsica should include a more representative set of species of the genus Armeria in order to more accurately determine the position of studied species. As a whole, several outgoups and a representative sample from the taxon under study (in this case, the Plagiobasis section) are preferred.
5. Why are specimens that do not combine with a clade of their own species not considered as hybrids? It would be better to separate specimens which belong to “pure” species and hybrids, and mark hybrid samples on multidimensional analysis graphs.
6. The study demonstrates that the morphological characteristics of seeds do not allow separating the species in the study group. The division into two clusters does not correspond to species or groups of species. Perhaps it would be better to use the characters of seed micromorphology (surface sculpture and ultrastructure).
7. The authors did not provide the information about the crossing system of studied species. It can be supposed that the majority of species are allogamous, because the authors mentioned hybridization and reticulate evolution in the group. According to the results presented in the manuscript, we can see signs of gene flow among “species”. We also see the absence of interspecific morphological boundaries. Therefore, it is unclear to the reader what kind of species concept the authors adhere to and why all the taxa studied are not considered as subspecies or varieties of one large species. The latter concept is also supported by a single chromosome number 2n=18.
As a whole, Armeria is a difficult genus to study, due to frequent hybridization and introgression among its species, as well as the concerted evolution of nrITS (Nieto Feliner et al., 2004). However, a more balanced and well-founded taxonomic decision should be made on the studied group of Armeria species. Morphometric analyses should be more convincing. On the basis of morphological and molecular data in the dataset, it is worth identifying hybrid samples and separating them from the "pure" representatives of the studied species.
Several small comments are maded in the manuscript.

Author Response
Dear Reviewer 3,
We wanted to express our heartfelt gratitude for the invaluable tips and guidance you provided during the review process. Your insightful suggestions have played a pivotal role in elevating the quality of our work. We truly appreciate the time and effort you dedicated to reviewing our manuscript and offering thoughtful feedback. Your contributions have had a profound impact on the overall quality of the ms.
We addressed each point raised in your review within our cover letter.
With warm regards,
The Authors

Round 2
Reviewer 3 Report
The authors of the article do not want to change their taxonomic views. I can agree with some of their arguments, but not all of them.
For example, the authors wrote: “Both A. sardoa and A. morisii are well distinguished both at phylogenetic and morphological level”. – This is not true. A.morisii is paraphyletic in relation to A. sulcitana (SP1, SP2, SP3). Other A. sulcitana samples (GO1, GO2, GO3) take more basal position on the tree. A. sardoa forms a clade, but this clade includes also two samples of A. leucocephala subsp. pubescens.
Another example. I recommended including a broader group of Armeria species in the phylogenetic analysis, because if you don't, you may get the wrong phylogeny. If you include a more representative set of species of the genus Armeria, it is possible that the species under study (A. leucocephala, A. sardoa, A. morisii, A. soleirolii and A. sulcitana) may mix with species from other taxonomic groups that are not common in Corsica and Sardinia. That's why using a single external group is not enough in most cases.
Nevertheless, the article can be published in its current form, as it contains many informative results that are useful for understanding the taxonomy and evolution of Armeria species in Corsica and Sardinia.